# ACHIEVING STRONG REGULARIZATION FOR DEEP NEURAL NETWORKS

## ABSTRACT

L1 and L2 regularizers are critical tools in machine learning due to their ability to simplify solutions. However, imposing strong L1 or L2 regularization with gradient descent method easily fails, and this limits the generalization ability of the underlying neural networks. To understand this phenomenon, we investigate how and why learning fails for strong regularization. Specifically, we examine how gradients change over time for different regularization strength and provide an analysis why the gradients diminish so fast when strong regularization is imposed. We find that there exists a tolerance level of regularization strength, where the learning completely fails if the regularization strength goes beyond it. We propose a simple but novel method, Delayed Strong Regularization, in order to moderate the tolerance level. Experiment results show that our proposed approach indeed achieves strong regularization for both L1 and L2 regularizers and improves both accuracy and sparsity on public data sets. Our source code is published.[1]

## 1 INTRODUCTION

Regularization has been very common for machine learning to prevent over-fitting and to obtain sparse solutions. Deep neural networks (DNNs), which have shown huge success in many tasks such as computer vision (Krizhevsky et al., 2012; Simonyan & Zisserman, 2014; He et al., 2016) and speech recognition (Hinton et al., 2012), often contain a number of parameters in multiple layers with non-linear activation functions, in order to gain enough expressive power. However, DNNs with many parameters are often prone to over-fitting, so the need for regularization has been emphasized. While new regularization techniques such as dropout (Srivastava et al., 2014) and pruning (Han et al., 2015) have been proposed to solve the problem, the traditional regularization techniques using L1 or L2 norms have cooperated with them to further improve the performance significantly. L1 regularization, often called Lasso (Tibshirani, 1996), obtains sparse solutions so that the required memory and power consumption are reduced while keeping reasonable accuracy. On the other hand, L2 regularization smooths the parameter distribution and reduces the magnitude of parameters, so the resulting solution is simple (*i.e.,* less prone to over-fitting) and effective. Indeed, our empirical results show that applying strong L2 regularization to the deep neural networks that already has dropout layers can reduce the error rate by up to 24% on a public data set.

Strong regularization is especially desired when the model contains too many parameters for the given amount of training data. This is often the case for deep learning tasks in practice because DNNs often contain millions of parameters while labeled training data set is limited and expensive. However, imposing strong L1 or L2 regularization on DNNs is difficult for gradient descent method due to the vanishing gradient problem. If we impose too strong regularization, the gradient from regularization becomes dominant, and DNNs stop learning. In this paper, we first study the interesting phenomenon that strong regularization fails in learning. We also provide an analysis why the gradients diminish so quickly that learning completely fails. Then, we propose a simple yet effective solution, Delayed Strong Regularization, which carries a time-dependent schedule of regularization strength. We find that we can overcome the failure in learning by waiting for the model to reach an "active learning" phase, where the gradients' magnitudes are significant, and then enforcing strong regularization. Delayed Strong Regularization enables us to obtain the superior performance that is otherwise hidden by learning failure in deep networks. The proposed approach is general and

---

[1] https://github.com/(anonymized)

does not require any additional computation. The experiment results indicate that the proposed approach indeed achieves strong regularization, consistently yielding even higher accuracy and higher compression rate that could not be achieved.

## 2 PROBLEM ANALYSIS

### 2.1 BACKGROUND

Let us denote a generic DNN by $\mathbf{y} = f(\mathbf{x}; \mathbf{w})$ where $\mathbf{x} \in \mathbb{R}^d$ is an input vector, $\mathbf{w} \in \mathbb{R}^n$ is a flattened vector of all parameters in the network $f$, and $\mathbf{y} \in \mathbb{R}^c$ is an output vector after feed-forwarding $\mathbf{x}$ through multiple layers in $f$. The network $f$ is trained by finding optimal set of $\mathbf{w}$ by minimizing the cost function within the training data $\{\mathbf{x}_i, \mathbf{d}_i\}_{i=1}^m$ as follows.

$$\mathbf{w}^* = \arg\min_{\mathbf{w}} \frac{1}{m} \sum_{i=1}^{m} \mathcal{L}(f(\mathbf{x}_i; \mathbf{w}), \mathbf{d}_i) + \lambda\Omega(\mathbf{w}) \tag{1}$$

where $\mathcal{L}$ is the loss function, which is usually cross-entropy loss for classification tasks. Here, the regularization term $\lambda\Omega(\mathbf{w})$ is added to simplify the solution, and $\lambda$ is set to zero for non-regularized cost function. A higher value of $\lambda$ means that stronger regularization is imposed.

The most commonly used regularization function is a squared L2 norm: $\Omega(\mathbf{w}) = ||\mathbf{w}||_2^2$, which is also called as *weight decay* in deep learning literature. This L2 regularizer has an effect of reducing the magnitude of the parameters $\mathbf{w}$, and the simpler solution becomes less prone to over-fitting. On the other hand, the L1 regularizer $\Omega(\mathbf{w}) = ||\mathbf{w}||_1$ is often employed to induce sparsity in the model (*i.e.,* make a portion of $\mathbf{w}$ zero). The sparse solution is often preferred to reduce computation time and memory consumption for deep learning since DNNs often require heavy computation and big memory space.

With the gradient descent method, each model parameter at time $t$, $\mathbf{w}_i^{(t)}$, is updated with the following formula:

$$\mathbf{w}_i^{(t+1)} = \mathbf{w}_i^{(t)} - \alpha \left( \frac{\partial \mathcal{L}}{\partial \mathbf{w}_i} + \lambda \frac{\partial \Omega}{\partial \mathbf{w}_i} \right)\bigg|_{\mathbf{w}_i = \mathbf{w}_i^{(t)}}$$

$$\frac{\partial \Omega}{\partial \mathbf{w}_i} = \begin{cases} 2 \times \mathbf{w}_i^{(t)}, & \text{if } \Omega(\mathbf{w}) = ||\mathbf{w}||_2^2 \\ 1 \times \text{sign}(\mathbf{w}_i^{(t)}), & \text{if } \Omega(\mathbf{w}) = ||\mathbf{w}||_1 \text{ and } \mathbf{w}_i^{(t)} \neq 0 \end{cases} \tag{2}$$

where $\alpha$ is a learning rate. As L1 norm is not differentiable at 0, the formula doesn't have value when $\mathbf{w}_i^{(t)} = 0$, but in practice, the subgradient 0 is often used. Please see Section 2.3 for more details. From the formula, we can see that L2 regularizer continuously reduces the magnitude of a parameter proportionally to it while L1 regularizer reduces the magnitude by a constant. In both regularizers, strong regularization thus means greatly reducing the magnitude of parameters.

### 2.2 IMPOSING STRONG REGULARIZATION MAKES LEARNING FAIL.

Strong regularization is especially useful for deep learning because the DNNs often contain a large number of parameters while the training data is limited in practice. However, we have observed a phenomenon where learning suddenly fails when strong regularization is imposed for gradient descent method, which is the most commonly used solver for deep learning. The example of the phenomenon is depicted in Figure 1. In the example, the architectures VGG-16 (Simonyan & Zisserman, 2014) and AlexNet (Krizhevsky et al., 2012) were employed for the data set CIFAR-100 (Krizhevsky & Hinton, 2009).[2] As shown, the accuracy increases as we enforce more regularization. However, it suddenly drops to 1.0% after enforcing a little more regularization, which means that the model entirely fails to learn. The depicted training loss also indicates that it indeed learns faster with stronger regularization ($\lambda = 1 \times 10^{-3}$), but the training loss does not improve at all when even stronger regularization is imposed ($\lambda = 2 \times 10^{-3}$).

In order to look at this phenomenon in more detail, we show how gradients and their proportion change in Figure 2. As depicted in Figure 2a, a model with moderate L2 regularization ($\lambda = 1 \times$

---

[2]Details of the experiment settings are described in Section 3

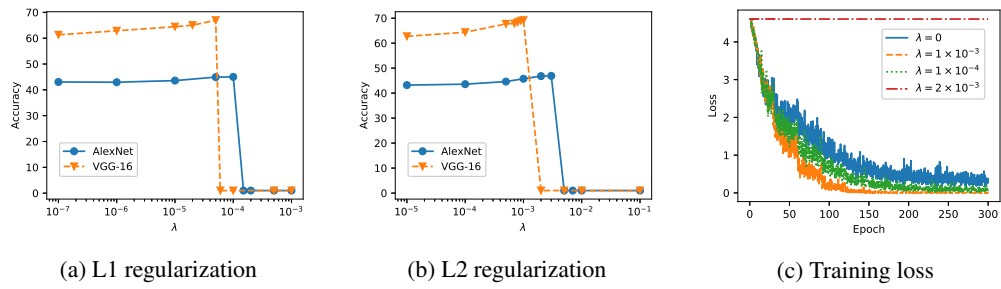

(a) L1 regularization    (b) L2 regularization    (c) Training loss

Figure 1: (a,b) Validation accuracies for different $\lambda$ when L1 and L2 regularization is applied to VGG-16 and AlexNet on the CIFAR-100 data set. Note the sharp accuracy drop. (c) Training loss when different $\lambda$ for L2 regularization is used for VGG-16 on CIFAR-100. Note that the regularization loss is excluded from the training loss. Best shown in color.

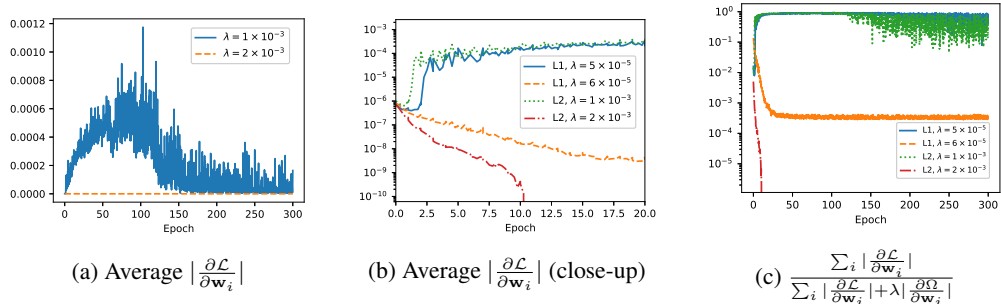

(a) Average $|\frac{\partial \mathcal{L}}{\partial \mathbf{w}_i}|$    (b) Average $|\frac{\partial \mathcal{L}}{\partial \mathbf{w}_i}|$ (close-up)    (c) $\frac{\sum_i |\frac{\partial \mathcal{L}}{\partial \mathbf{w}_i}|}{\sum_i |\frac{\partial \mathcal{L}}{\partial \mathbf{w}_i}| + \lambda |\frac{\partial \Omega}{\partial \mathbf{w}_i}|}$

Figure 2: Gradients for different $\lambda$ by VGG-16 on CIFAR-100. (a) Average amount of gradient from $\mathcal{L}$ when L2 regularization is applied. (b) A close-up version of (a) with y-axis in log-scale. (c) The proportion of gradients from $\mathcal{L}$ to all gradients. Best shown in color.

$10^{-3}$) follows a path that has a relatively steep slope during the first 150 epoch, and then it converges with a gentle slope. However, a model with a little stronger L2 regularization ($\lambda = 2 \times 10^{-3}$) does not follow a path that has a good slope, so it does not really have a chance to learn from gradients. A close-up view of this in the first 20 epochs is depicted in Figure 2b. The models with moderate L1 and L2 regularization seem to follow a good path in a couple of epochs. Through following the good path, the models keep the proportion of the gradients from $\mathcal{L}$ to all gradients dominant, especially for the first 150 epochs (Figure 2c). On the other hand, the models with a little stronger regularization fail to follow such path and the gradients from $\mathcal{L}$ decrease exponentially (Figure 2b). Since the magnitude of gradients from $\mathcal{L}$ decreases faster than that from $\Omega$, the proportion of the latter to all gradients becomes dominant (Figure 2c), and it results in failure in learning. From this observation, we can see that there exists a *tolerance level* of regularization strength, which decides success or failure of entire learning.

**Why does the magnitude of the gradient from $\mathcal{L}$ decrease so fast?**    It is not difficult to see why the magnitude of $\frac{\partial \mathcal{L}}{\partial \mathbf{w}_i}$ decreases so fast when the regularization is strong. In deep neural networks, the gradients are dictated by back-propagation. It is well known that the gradients at the $l^{\text{th}}$ layer are given by

$$\frac{\partial \mathcal{L}}{\partial \mathbf{w}^{(l)}} = \boldsymbol{\delta}^{(l)} \left( \mathbf{a}^{(l-1)} \right)^T \tag{3}$$

where $\mathbf{a}^{(l-1)}$ is the output of the neurons at the $(l-1)^{\text{th}}$ layer and $\boldsymbol{\delta}^{(l)}$ is the $l^{\text{th}}$-layer residual which follows the recursive relation

$$\boldsymbol{\delta}^{(l)} = \left( \mathbf{w}^{(l+1)} \right)^T \boldsymbol{\delta}^{(l+1)} \odot \mathbf{a}'^{(l)} \tag{4}$$

where $\odot$ and $\mathbf{a}'$ denote the element-wise multiplications and derivatives of the activation function respectively.

Using the recursive relation, we obtain

$$
\begin{aligned}
\frac{\partial \mathcal{L}}{\partial \, \mathbf{w}^{(l)}} \;\;=\;\; & ( \, \mathbf{w}^{(l+1)} \, )^T \, ( \, \mathbf{w}^{(l+2)} \, )^T \, \cdots \, ( \, \mathbf{w}^{(L)} \, )^T \, \boldsymbol{\delta}^{(L)} \\
& \odot \; \mathbf{a}'^{(L-1)} \odot \mathbf{a}'^{(L-2)} \odot \cdots \odot \mathbf{a}'^{(l+1)} \odot \mathbf{a}'^{(l)} \, ( \, \mathbf{a}^{(l-1)} \, )^T
\end{aligned}
\tag{5}
$$

If the regularization is too strong, the weights would be significantly suppressed as shown in Figure 5b. From (5), since the gradients are proportional to the product of the weights at later layers (whose magnitudes are typically much less than 1 for strong regularization), they are even more suppressed.

In fact, the suppression is more severe than what we have deduced above. The factor $\mathbf{a}^{(l-1)}$ in (5) could actually lead to further suppression to the gradients when the weights are very small, for the following reasons. First of all, we use ReLU as the activation function and it could be written as

$$
\mathrm{ReLU}(x) = x \, \Theta(x)
\tag{6}
$$

where $\Theta(x)$ is the Heaviside step function. Using this, we could write

$$
\mathbf{a}^{(l-1)} = \left( \mathbf{w}^{(l-1)} \, \mathbf{a}^{(l-2)} \right) \odot \Theta \left( \mathbf{w}^{(l-1)} \, \mathbf{a}^{(l-2)} \right)
\tag{7}
$$

Applying (7) recursively, we can see that $\mathbf{a}^{(l-1)}$ is proportional to the product of the weights at previous layers. Again, when the weights are suppressed by strong regularization, $\mathbf{a}^{(l-1)}$ would be suppressed correspondingly. Putting everything together, we can conclude that in the presence of strong regularization, the gradients are far more suppressed than the weights.

Strictly speaking, the derivations above are valid only for fully-connected layers. For convolutional layers, the derivations are more complicated but similar. Our conclusions above would still be valid.

**Normalization** Normalization techniques such as batch normalization (Ioffe & Szegedy, 2015) and weight normalization (Salimans & Kingma, 2016) can be possible approaches to prevent the $\mathcal{L}$ gradients from diminishing quickly. However, it has been shown that L2 regularization has no regularizing effect when combined with normalization but only influences on the effective learning rate, resulting in good performance (van Laarhoven, 2017). In other words, the normalization techniques do not really simplify the solution as the decrease of parameter magnitude is canceled by normalization. This does not meet our goal, which is to heavily simplify solutions to reduce over-fitting, so we propose an approach that meets our goal.

### 2.3 Our Proposed Approach: Delayed Strong Regularization

Since we have seen that stronger regularization can result in better performance in Figure 1, we propose a method that is able to accommodate strong regularization. Specifically, we introduce a time-dependent regularization strength, $\lambda_t$, to the equation (2), and it is defined as

$$
\lambda_t = \begin{cases} 0 & \text{if epoch}(t) < \gamma \\ \lambda & \text{otherwise} \end{cases}
\tag{8}
$$

where epoch$(t)$ gets the epoch number of the time step $t$, and $\gamma$ is a hyper-parameter that is set through cross-validation. The formula means that we do not impose any regularization until $\gamma^{\text{th}}$ epoch, and then impose the strong regularization in each training step. The underlying hypothesis is that once the model follows a good learning path, $i.e.$, the gradient from $\mathcal{L}$ is big enough, it won't easily change its direction because of the steep slope, and thus, it can learn without failure. We empirically verify our hypothesis in the experiment section. The hyper-parameter $\gamma$ is relatively easy to set because the models often follow the good path in a couple of epochs, and once they follow such path, learning does not fail. We recommend using $2 \leq \gamma \leq 20$. Please note that our approach is different from imposing a slightly weaker regularization throughout the whole training. The reduced amount by not skipping regularization for the first few epochs is negligible compared to the total reduced amount by regularization. In addition, we empirically show that our approach can achieve a much higher sparsity than the baseline in the parameter space.

The proposed method is easy to implement, and the hyper-parameter is easy to set. Also, the method is very close to the traditional regularization method so that it inherits the traditional one's good

performance for non-strong regularization while it also achieves strong regularization. Although the method is very simple, we found that it shows the best accuracy among the approaches we tried in our preliminary experiments while it is the simplest. The preliminary experiments are further discussed in Appendix B.

**Proximal gradient algorithm for L1 regularizer**    Meanwhile, since L1 norm is not differentiable at zero, we employ the proximal gradient algorithm (Parikh et al., 2014), which enables us to obtain proper sparsity (*i.e.,* guaranteed convergence) for non-smooth regularizers. We use the following update formulae:

$$
\mathbf{w}_i^{(t')} = \mathbf{w}_i^{(t)} - \alpha \left. \frac{\partial \mathcal{L}}{\partial \mathbf{w}_i} \right|_{\mathbf{w}_i = \mathbf{w}_i^{(t)}}
$$

$$
\mathbf{w}_i^{(t+1)} = \mathrm{prox}_{\alpha\lambda\Omega}(\mathbf{w}_i^{(t')}) = S(\mathbf{w}_i^{(t')}, \alpha\lambda)
$$

$$
S(a, z) = \begin{cases} a - z & \text{if } a > z \\ a + z & \text{if } a < -z \\ 0 & \text{otherwise} \end{cases}
\tag{9}
$$

where $S$ is a soft-thresholding operator. Basically, the algorithm assigns zero to a parameter if its next updated value is smaller than $\alpha\lambda$. In other cases, it just decreases the magnitude of the parameter as usual.

## 3   EXPERIMENTS

We first evaluate the effectiveness of our proposed method with popular architectures, AlexNet (Krizhevsky et al., 2012) and VGG-16 (Simonyan & Zisserman, 2014) on the public data sets CIFAR-10 and CIFAR-100 (Krizhevsky & Hinton, 2009). Then, we employ variations of VGG on another public data set SVHN (Netzer et al., 2011), in order to see the effect of the number of hidden layers on the tolerance level. Please note that we do not employ architectures that contain normalization techniques such as batch normalization (Ioffe & Szegedy, 2015), for the reason described in Section 2.2. The data set statistics are described in Table 1. VGG-11 and VGG-19 for SVHN contain 9.8 and 20.6 millions of parameters.

Table 1: Data set statistics for CIFAR-10 and CIFAR-100.

| Data Set | Image Resolution | Classes | Training Images per Class | Test Images per Class | AlexNet Parameters | VGG-16 Parameters |
|---|---|---|---|---|---|---|
| CIFAR-10 | 32×32 | 10 | 5000 | 1000 | 2.6M | 15.2M |
| CIFAR-100 | 32×32 | 100 | 500 | 100 | 2.6M | 15.3M |
| SVHN | 32×32 | 10 | 7325.7 (avg.) | 2603.2 (avg.) | - | 15.2M |

Regularization is applied to all network parameters except bias terms. We use PyTorch[3] framework for all experiments, and we use its official computer vision library[4] for the implementations of the networks. In order to accommodate the data sets, we made some modifications to the networks. The kernel size of AlexNet's max-pooling layers is changed from 3 to 2, and the first convolution layer's padding size is changed from 2 to 5. All of its fully connected layers are modified to have 256 neurons. For VGG, we modified the fully connected layers to have 512 neurons. The output layers of both networks have 10 neurons for CIFAR-10 and SVHN, and 100 neurons for CIFAR-100. The networks are learned by stochastic gradient descent algorithm with momentum of 0.9. The parameters are initialized according to He et al. (2015). The batch size is set to 128, and the initial learning rate is set to 0.05 and decays by a factor of 2 every 30 epochs during the whole 300-epoch training. In all experiments, we set $\gamma = 5$. We did not find significantly different results for $2 \leq \gamma \leq 20$. Please note that we still use drop out layers (with drop probability 0.5) and pre-

---

[3] http://pytorch.org/
[4] https://github.com/pytorch/vision

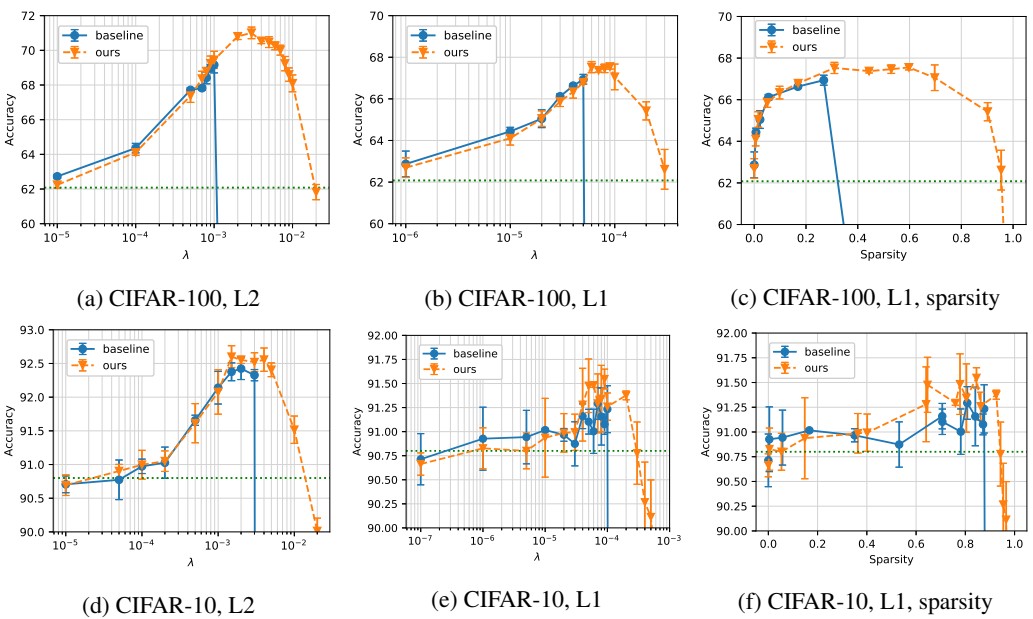

Figure 3: Accuracies obtained by VGG-16 with L1 and L2 regularization on CIFAR-100 (a,b,c) and CIFAR-10 (d,e,f). A green dotted horizontal line is an accuracy obtained by a model without L1/L2 regularization (but with dropout). Accuracy for different sparsity is shown in (c) and (f). The error bars indicate 95% confidence interval.

process training data[5] in order to report the extra performance boost on top of common regularization techniques.

AlexNet and VGG-16 are experimented for different regularization methods (L1 and L2) and different data sets (CIFAR-10 and CIFAR-100), yielding 8 combinations of experiment sets. Then, VGG-11, VGG-16, and VGG-19 are experimented for L1 and L2 regularization methods on SVHN, yielding 6 experiment sets. For each experiment set, we set the **baseline** method as the one with well-tuned L1 or L2 regularization but without our time-dependent regularization strength. For each regularization, we try more than 10 different values of $\lambda$, and for each value, we report average accuracy of three independent runs and report 95% confidence interval. We perform statistical significance test (t-test) for the improvement over the baseline method and report the p-value. We also report **sparsity** of each trained model, which is the proportion of the number of zero-valued parameters to the number of all parameters. Please note that we mean the sparsity by the one actually derived by the models, not by pruning parameters with threshold after training.

### 3.1 EXPERIMENT RESULTS ON CIFAR-10 AND CIFAR-100

The experiment results by VGG-16 are depicted in Figure 3. As we investigated in Section 2.2, the baseline method suddenly fails beyond certain values of tolerance level. However, our proposed method does not fail for higher values of $\lambda$. As a result, our model can achieve higher accuracy as well as higher sparsity. In practice, L2 regularization is used more often than L1 regularization due to its superior performance, and this is true for our VGG-16 experiments too. Using L2 regularization, our model improves the model without L1 or L2 regularization but with dropout, by 14.4% in accuracy, which is about 24% of error rate improvement. Tuning L2 regularization parameter is difficult as the curves have somewhat sharp peak, but our proposed method ease the problem to some extent by preventing the sharp drop. Our L1 regularizer obtains much better sparsity for the similar level of accuracy (Figure 3c), which means that strong regularization plays an important role in compressing neural networks. The improvement is more prominent on CIFAR-100 than on

---

[5]We apply horizontal random flipping and random cropping to original images in each batch. We do not apply them to SVHN as they may harm the performance.

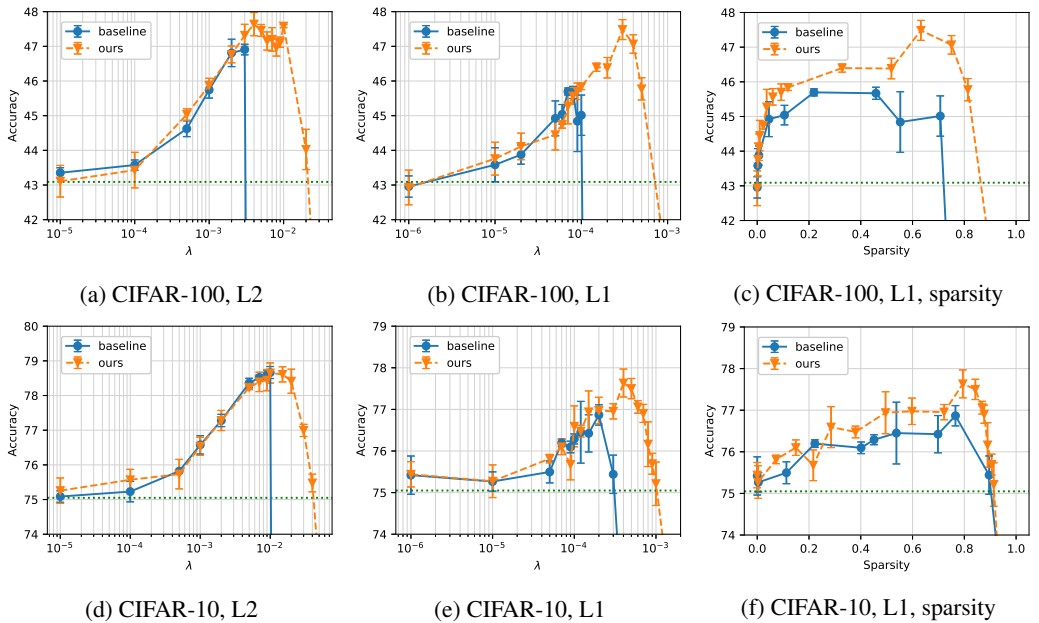

(a) CIFAR-100, L2    (b) CIFAR-100, L1    (c) CIFAR-100, L1, sparsity

(d) CIFAR-10, L2    (e) CIFAR-10, L1    (f) CIFAR-10, L1, sparsity

Figure 4: Accuracies obtained by AlexNet with L1 and L2 regularization on CIFAR-100 (a,b,c) and CIFAR-10 (d,e,f). A green dotted horizontal line is an accuracy obtained by a model without L1/L2 regularization (but with dropout). Accuracy for different sparsity is shown in (c) and (f). The error bars indicate 95% confidence interval.

CIFAR-10, and we think this is because over-fitting can more likely occur on CIFAR-100 as there are less images per class than on CIFAR-10.

The experiment results by AlexNet are depicted in Figure 4. Again, our proposed method achieves higher accuracy and sparsity in general. Unlike VGG-16, we obtain more improvement over baseline with L1 regularization than with L2 regularization. In addition, the curves make sharper peaks than those by VGG-16 especially for the sparsity regularizer (L1).

Interestingly, our proposed method often obtains higher accuracy even when the baseline does not fail on CIFAR-10, and this is only prominent when the regularization strength is relatively strong (better shown in Figure 3f, 4c, 4f). This may be because avoiding strong regularization in the early stage of training can help the model to explore more spaces freely, and the better exploration results in finding superior local optima.

The overall experiment results are shown in Table 2. It shows that there is more performance improvement by L1/L2 regularization on VGG-16 than on AlexNet, which is reasonable since VGG-16 contains about 6 times more parameters so that it is more prone to over-fitting. Our proposed model always improves the baselines by up to 3.89%, except AlexNet with L1 regularization on CIFAR-10, and most (6 out of 7) improvements are statistically significant ($p<0.05$). Our L1 regularization models always obtain higher sparsity with compression rate up to $4.2\times$ than baselines, meaning that our model is promising for compressing neural networks.

We also show in Figure 5 how gradients and weights change when our method and the baseline are applied. We hypothesized that if the model reaches an "active learning" phase with an elevated gradient amount, it does not suffer from vanishing gradients any more even when strong regularization is enforced. The Figure 5a shows that our model indeed reaches there by skipping strong regularization for the first five epochs, and and it keeps learning even after strong regularization is enforced. In Figure 5b, although the same strong regularization is enforced since epoch 5, the magnitude of weights in our model stops decreasing around epoch 20, while that in baseline (green dotted line) keeps decreasing towards zero. This means that our model can cope with strong regularization, and it maintains its equilibrium between gradients from $\mathcal{L}$ and those from regularization.

Table 2: Overall experiment results. The 95% confidence interval is reported after $\pm$ symbol. The improvement is calculated by comparing with baseline methods. Compression rate is calculated by how much the regularization reduces the number of non-zero-valued parameters while having an accuracy similar to L1 baseline. Note that when no L1/L2 regularization is imposed, dropout is still employed.

| | Model | CIFAR-100 | | CIFAR-10 | |
|---|---|---|---|---|---|
| | | Sparsity | Acc. (%) | Sparsity | Acc. (%) |
| VGG-16 | No L1/L2 | 0 | 62.08 ± 0.81 | 0 | 90.80 ± 0.23 |
| | L2 baseline | 0 | 69.16 ± 0.46 | 0 | 92.42 ± 0.16 |
| | L2 ours | 0 | 71.01 ± 0.33 | 0 | 92.60 ± 0.16 |
| | Acc. improvement | - | +2.67% (p=0.002) | - | +0.19% (p=0.103) |
| | L1 baseline | 0.269 | 66.94 ± 0.24 | 0.808 | 91.29 ± 0.16 |
| | L1 ours | 0.303 | 67.55 ± 0.12 | 0.845 | 91.55 ± 0.10 |
| | Acc. improvement | - | +0.91% (p=0.011) | - | +0.28% (p=0.037) |
| | L1 ours (sparse) | 0.697 | 67.06 ± 0.62 | 0.926 | 91.38 ± 0.05 |
| | Compression rate over L1 baseline | 2.4× | - | 2.6× | - |
| AlexNet | No L1/L2 | 0 | 43.09 ± 0.25 | 0 | 75.05 ± 0.20 |
| | L2 baseline | 0 | 46.91 ± 0.15 | 0 | 78.66 ± 0.17 |
| | L2 ours | 0 | 47.64 ± 0.33 | 0 | 78.65 ± 0.29 |
| | Acc. improvement | - | +1.56% (p=0.017) | - | −0.01% |
| | L1 baseline | 0.219 | 45.70 ± 0.10 | 0.766 | 76.87 ± 0.24 |
| | L1 ours | 0.632 | 47.48 ± 0.29 | 0.794 | 77.63 ± 0.34 |
| | Acc. improvement | - | +3.89% (p=0.002) | - | +0.99% (p=0.013) |
| | L1 ours (sparse) | 0.814 | 45.77 ± 0.32 | 0.877 | 76.90 ± 0.22 |
| | Compression rate over L1 baseline | 4.2× | - | 1.9× | - |

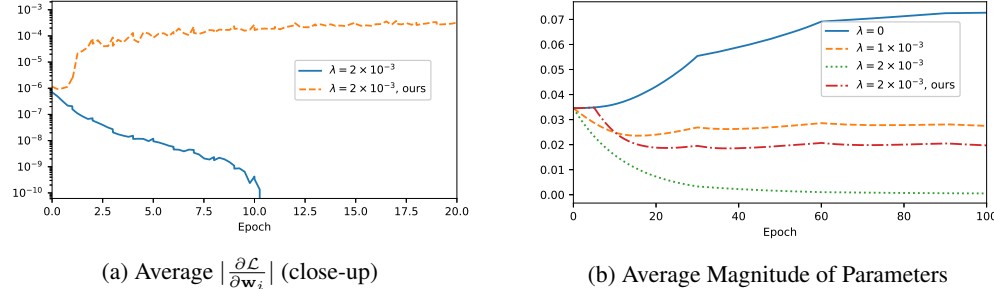

(a) Average $\left|\frac{\partial \mathcal{L}}{\partial \mathbf{w}_i}\right|$ (close-up)    (b) Average Magnitude of Parameters

Figure 5: Experiment results by L2 regularization with VGG-16 on CIFAR-100. The results are from baseline method unless it is labeled as "ours". Our proposed Delayed Strong Regularization does not fail in learning.

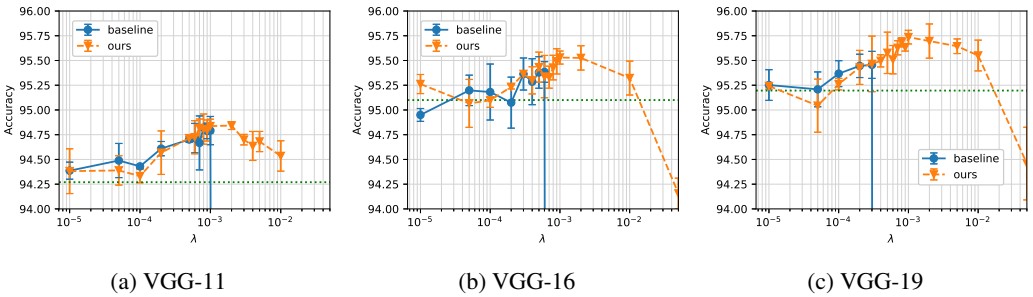

(a) VGG-11        (b) VGG-16        (c) VGG-19

Figure 6: Accuracies obtained by variations of VGG with L2 regularization on SVHN. A green dotted horizontal line is an accuracy obtained by a model without L2 regularization (but with dropout). The error bars indicate 95% confidence interval.

Table 3: Experiment results on SVHN. The improvement is calculated by comparing with baseline methods. Note that when no L1/L2 regularization is imposed, dropout is still employed.

| Model | VGG-11 | | VGG-16 | | VGG-19 | |
|---|---|---|---|---|---|---|
| | Sparsity | Acc. (%) | Sparsity | Acc. (%) | Sparsity | Acc. (%) |
| No L1/L2 | 0 | $94.27 \pm 0.13$ | 0 | $95.10 \pm 0.22$ | 0 | $95.20 \pm 0.10$ |
| L2 baseline | 0 | $94.82 \pm 0.02$ | 0 | $95.38 \pm 0.10$ | 0 | $95.45 \pm 0.14$ |
| L2 ours | 0 | $94.84 \pm 0.04$ | 0 | $95.53 \pm 0.07$ | 0 | $95.74 \pm 0.07$ |
| Acc. improvement | - | +0.02% (p=0.200) | - | +0.16% (p=0.047) | - | +0.30% (p=0.019) |
| L1 baseline | 0.519 | $94.68 \pm 0.08$ | 0.450 | $95.34 \pm 0.11$ | 0.122 | $95.37 \pm 0.11$ |
| L1 ours | 0.621 | $94.78 \pm 0.07$ | 0.795 | $95.38 \pm 0.11$ | 0.518 | $95.50 \pm 0.15$ |
| Acc. improvement | - | +0.11% (p=0.056) | - | +0.04% (p=0.324) | - | +0.13% (p=0.180) |
| L1 ours (sparse) | 0.845 | $94.71 \pm 0.01$ | 0.795 | $95.38 \pm 0.11$ | 0.911 | $95.41 \pm 0.07$ |
| Compression rate over L1 baseline | 3.1× | - | 2.7× | - | 9.9× | - |

## 3.2 EXPERIMENT RESULTS ON SVHN

The analysis in Section 2.2 implies that the number of hidden layers would affect the tolerance level when strong regularization is imposed. That is, if there are more hidden layers in the neural network architecture, the learning will fail more easily by strong regularization. In order to check the hypothesis empirically, we employ variations of the VGG architecture, *i.e.,* VGG-11, VGG-16, and VGG-19, which contain 11, 16, and 19 hidden layers, respectively. We experiment them on the SVHN data set.

The results by L2 regularization are depicted in Figure 6. As shown, the peak of our method's performance is formed around $\lambda = 1 \times 10^{-3}$. As more hidden layers are added to the network, the tolerance level where the performance suddenly drops by the baseline is shifted to left, as hypothesized by our analysis. The results by L1 regularization are in Appendix A, and it is shown that VGG-19 more easily fails as the parameters become more sparse. The overall experiment results are shown in Table 3. As the method without L1/L2 regularization already performs well on this data set and there are relatively many training images per class, the improvement by L1/L2 regularization is not big. Our method still outperforms the baseline in all experiments (6 out of 6), but the improvement is less statistically significant compared to CIFAR-10 and CIFAR-100 data sets. The compression rate is especially good for VGG-19 mainly because its tolerance level is low so that the baseline can only achieve low sparsity.

## 4 RELATED WORK

The related work is partially covered in Section 1, and we extend other related work here. It has been shown that L2 regularization is important for training DNNs (Krizhevsky et al., 2012; Deng et al., 2013). Although there has been a new regularization method such as dropout, L2 regularization has been shown to reduce the test error effectively when combined with dropout (Srivastava et al., 2014). Meanwhile, L1 regularization has also been used often in order to obtain sparse solutions. To reduce computation and power consumption, L1 regularization and its variations such as group sparsity

regularization has been promising for deep neural networks (Wen et al., 2016; Scardapane et al., 2017; Yoon & Hwang, 2017). However, for both L1 and L2 regularization, the phenomenon that learning fails with strong regularization has not been emphasized previously. Bergstra & Bengio (2012) showed that tuning hyper-parameters such as L2 regularization strength can be effectively done through random search instead of grid search, but they did not study how and why learning fails or how strong regularization can be successfully achieved. Yosinski et al. (2015) visualized activations to understand deep neural networks and showed that strong L2 regularization fails to learn. However, it was still not shown how and why learning fails and how strong regularization can be achieved. To the best of our knowledge, there is no existing work that is dedicated to studying the phenomenon that learning fails with strong regularization and to proposing a method that can avoid the failure.

## 5 DISCUSSION

In this work, we studied the problem of achieving strong regularization for deep neural networks. Strong regularization with gradient descent algorithm easily fails for deep neural networks, but few work addressed this phenomenon in detail. We provided investigation and analysis of the phenomenon, and we found that there is a strict tolerance level of regularization strength. To avoid this problem, we proposed a novel but simple method: Delayed Strong Regularization. We performed experiments with fine tuning of regularization strength. Evaluation results show that (1) our model successfully achieves strong regularization on deep neural networks, verifying our hypothesis that the model will keep learning once it reaches an "active learning" phase, (2) with strong regularization, our model obtains higher accuracy and sparsity, (3) the number of hidden layers in neural networks affects the tolerance level, and (4) L1/L2 regularization is difficult to tune, but it can yield great performance boost when tuned well.

There are limitations in this work. Our proposed method can be especially useful when strong regularization is desired. For example, deep learning projects that cannot afford a huge labeled data set can benefit from our method. However, strong regularization may not be necessary in some other cases where the large labeled data set is available or the networks do not contain many parameters. In addition, our experiments were not performed on a bigger data set such as ImageNet data set. We need to fine-tune the models with different regularization parameters, and we also need multiple training sessions of each model to obtain confidence interval. For example, the experiment results in Figure 3 and 4 include 750 training sessions in total. This is something we cannot afford with ImageNet data set, which requires several weeks of training for EACH session (unless we have GPU clusters). Our approach cannot be applied to architectures containing normalization techniques for the reason in Section 2.2. We actually tried to intentionally exclude normalization part from Residual Networks (He et al., 2016) and train the model to see if we can apply our method to non-normalized Residual Networks. However, we could not control the exploding gradients caused by the exclusion of normalization.

Our work can be further extended in several ways. Since our model can achieve strong regularization, it will be interesting to see how the strongly regularized model performs if combined with pruning-related methods (Han et al., 2015; 2016). We applied our approach to only L1 and L2 regularizers, but applying it to other regularizers such as group sparsity regularizers will be promising as they are often employed for DNNs to compress networks. Lastly, our proposed Delayed Strong Regularization is very simple, so one can easily extend it to more complicated methods. All these directions are left as our future work.

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

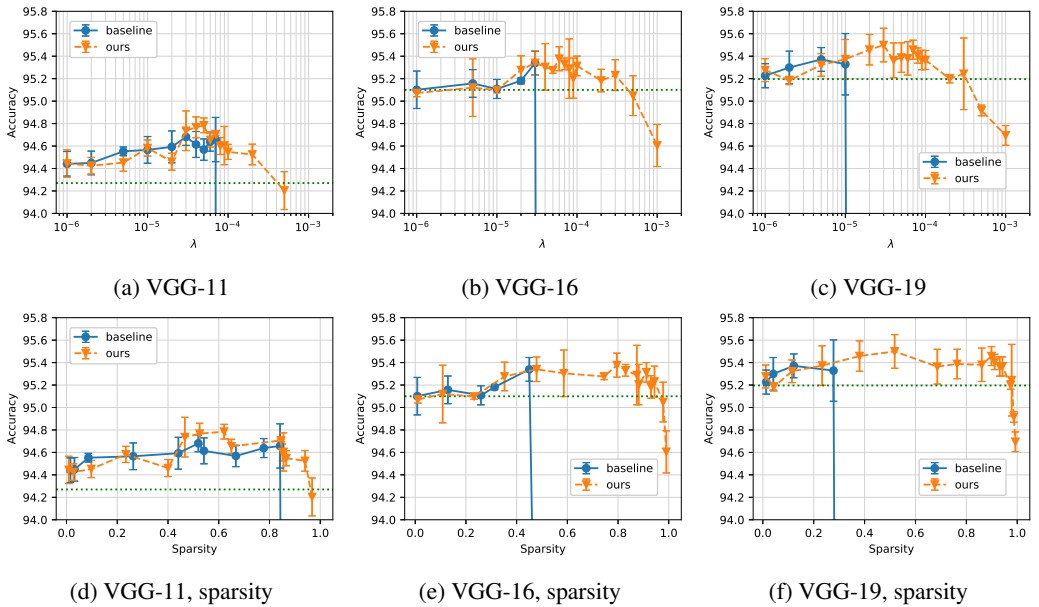

Figure 7: Accuracies obtained by variations of VGG with L1 regularization on SVHN. A green dotted horizontal line is an accuracy obtained by a model without L1 regularization (but with dropout). Accuracy for different sparsity is shown in (d), (e), and (f). The error bars indicate 95% confidence interval.

Jaehong Yoon and Sung Ju Hwang. Combined group and exclusive sparsity for deep neural networks. In *International Conference on Machine Learning*, pp. 3958–3966, 2017.

Jason Yosinski, Jeff Clune, Anh Nguyen, Thomas Fuchs, and Hod Lipson. Understanding neural networks through deep visualization. *arXiv preprint arXiv:1506.06579*, 2015.

# Appendices

## A    EXPERIMENT RESULTS ON SVHN BY L1 REGULARIZER

To empirically check the effect of the number of hidden layers on the tolerance level, we experimented variations of VGG on SVHN, and we showed the results by L2 regularizer in Section 3.2. Here, we show the results by L1 regularizer in Figure 7. As more hidden layers are included to the network, the tolerance level where the baseline method suddenly fails is shifted to left. Such pattern in baseline method is more clearly shown in the accuracy vs. sparsity plots. VGG-19 fails to learn even when it loses only 27% of its parameters, whereas VGG-11 can still learn after losing 84% of its parameters.

## B    PRELIMINARY EXPERIMENTS

The reason why we proposed a very simple method is that it is effective while it is simple to implement. The only additional hyper-parameter, which is the number of initial epochs to skip regularization, is also not difficult to set. We think that the proposed method is very similar to the traditional regularization method so that it inherits the traditional one's good performance for non-strong regularization while it also achieves strong regularization.

We actually tried a couple more approaches other than the proposed one in our preliminary experiments. We found that the proposed one shows the best accuracy among the approaches we tried

while it is the simplest. For example, we tried an approach that can be regarded as a warm-start strategy. It starts with the regularization parameter $\lambda_t = 0$, and then it gradually increases $\lambda_t$ to $\lambda$ for $\gamma$ epochs, where $\gamma >= 0$ and it is empirically set. We found that it can achieve strong regularization, but its best accuracy is similar to or slightly lower than that of our proposed approach.

We also tried a method that is similar to Ivanov regularization (Pelckmans et al., 2004). In this method, the regularization term is applied only when the L1 norm of the weights is greater than a certain threshold. To enforce strong regularization, we set $\lambda$ just above the tolerance level that is found by the baseline method. However, this method did not accomplish any learning. The reason is that, to reach the level of L1 norm that is low enough, the model needs to go through the strong regularization for the first few epochs, and the neurons already lose its learning ability during this period like the baseline method. If we set $\lambda$ below the tolerance level, it cannot reach the desired L1 norm without strong regularization, and thus the performance is inferior to our proposed method.

Meanwhile, an approach that applies strong regularization first and then continuously reduces the regularization strength is used in sparse learning for convex optimization. This approach is opposite to our approach in that ours avoids strong regularization for the first few epochs and then apply strong regularization afterwards. We performed a simple experiment with VGG-16 on CIFAR-100 to see if the approach can perform well for deep neural networks. We set the initial regularization parameter $\lambda = 2 \times 10^{-3}$ and $\lambda = 6 \times 10^{-5}$ for L2 and L1 regularization, respectively, which are just above the "tolerance level". Then, we continuously reduced $\lambda_t$ to zero throughout the training session. The trained models didn't show any improvement over "random guess", which means that they were not able to learn. Once the strong regularization is enforced in the beginning, the magnitudes of weights decrease quickly. This in turn drives the magnitudes of gradients to diminish exponentially in deep neural networks as explained in Section 2.2, and thus, the model loses its ability to learn after a short period of strong regularization.

