# OpenReview forum: "Achieving Strong Regularization for Deep Neural Networks"
_ICLR.cc/2018/Conference — Reject_

### Official Review · AnonReviewer3 · 2017-11-26
**The authors proposed to apply L1/L2 regularization in training of deep neural networks after 5-10 epochs.**

**Rating:** 6
**Confidence:** 2

**Review:**

The authors studied the behavior that a strong regularization parameter may lead to poor performance in training of deep neural networks. Experimental results on CIFAR-10 and CIFAR-100 were reported using AlexNet and VGG-16. The results seem to show that a delayed application of the regularization parameter leads to improved classification performance.

The proposed scheme, which delays the application of regularization parameter, seems to be in contrast of the continuation approach used in sparse learning. In the latter case, a stronger parameter is applied, followed by reduced regularization parameter. One may argue that the continuation approach is applied in the convex optimization case, while the one proposed in this paper is for non-convex optimization. It would be interesting to see whether deep networks can benefit from the continuation approach, and the strong regularization parameter may not be an issue because the regularization parameter decreases as the optimization progress goes on.

One limitation of the work, as pointed by the authors, is that experimental results on big data sets such as ImageNet is not reported.

---

> ### Author Response · Authors · 2018-01-03
> **Thank you for the interesting suggestion. Here's our answer and experiment result for the suggestion.**
>
> We thank the reviewer for the interesting suggestion.
>
> In our paper, we proposed to adopt strong regularization for two main goals. One goal is to improve the model's accuracy, and the other goal is to compress the model while the accuracy is kept at the same level. Your suggestion meets especially the latter one. However, we think that it will be very difficult for your suggested approach to perform well with deep neural networks. Once the strong regularization is enforced in the beginning, the magnitudes of weights decrease quickly. This in turn drives the magnitudes of gradients to diminish exponentially in deep neural networks as explained in Section 2.2, and thus, the model loses its ability to learn after a short period of strong regularization. Even if we reduce the strength of the regularization after the strong regularization, it will be difficult for the model to recover its learning ability because the gradients are proportional to the product of the weights at later layers.
>
> In order to actually check if your suggested method works, we performed a simple experiment with VGG-16 on CIFAR-100. We set the initial regularization parameter \lambda=2*10^-3 and 6*10^-5 for L2 and L1 regularization, respectively, which are just above the "tolerance level". Then, we continuously reduced \lambda_t to zero throughout the training session. The trained models didn't show any improvement over "random guess", which means that they were not able to learn.
>
> We could not perform experiments on ImageNet for the following reason (as answered to other reviewers).
>
> As described in Section 5, we did not experiment on ImageNet only because it requires much time to train each model although we need to train many models. We need to fine-tune the models with different regularization parameters, and we also need multiple training sessions of each model to obtain confidence interval. For example, the experiment results in Figure 3 and 4 include 750 training sessions in total. This is something we cannot afford with ImageNet data set, which requires several weeks of training for EACH session (unless we have GPU clusters).
>
> However, we instead performed additional experiments on another data set. Specifically, we will add results of different VGG architectures on the SVHN data set, in order to see the difference in the tolerance level that is caused by a different number of hidden layers. We will add these results in the revised version.
>
> We will make these points clear in the revised draft.

---

### Official Review · AnonReviewer2 · 2017-11-27
**The authors propose a modification on regularization weights in an attempt to impose very strong regularization on backpropataion. There are several issues with the proposal as well as its efficacy in practice. The current statues of the paper needs more work.**

**Rating:** 4
**Confidence:** 5

**Review:**

The paper is well motivated and written. However, there are several issues.
1. As the regularization constant increases, the performance first increases and then falls down -- this specific aspect is well known for constrained optimization problems. Further, the sudden drop in performance also follows from vanishing gradients problem in deep networks. The description for ReLUs in section 2.2 follows from these two arguments directly, hence not novel. Several of the key aspects here not addressed are:
1a. Is the time-delayed regularization equivalent to reducing the value (and there by bringing it back to the 'good' regime before the cliff in the example plots)?
1b. Why should we keep increasing the regularization constant beyond a limit? Is this for compressing the networks (for which there are alternate procedures), or anything else. In other words, for a non-convex problem (about whose landscape we know barely anything), if there are regimes of regularizers that work well (see point 2) -- why should we ask for more stronger regularizers? Is there any optimization-related motivation here (beyond the single argument that networks are overparameterized)?
2. The proposed experiments are not very conclusive. Firstly, the authors need to test with modern state-of-the-art architectures including inception and residual networks. Secondly, more datasets including imagenet needs to be tested. Unless these two are done, we cannot assertively say that the proposal seems to do interesting things. Thirdly, it is not clear what Figure 5 means in terms of goodness of learning. And lastly, although confidence intervals are reported for Figures 3,4 and Table 2, statistical tests needs to be performed to report p-values (so as to check if one model significantly beats the other).

---

> ### Author Response · Authors · 2018-01-03
> **We described why we do not employ the state-of-the-art architectures in the paper. Here is further clarification.**
>
> 2.  Clarification
>
> - State-of-the-art architectures
> As explained in Section 2.2 and 3, we do not employ architectures that contain normalization techniques. Please see the Normalization paragraph of Section 2.2 for details. Unfortunately, most recent architectures contain normalization techniques. In order to apply our approach to recent architectures such as Residual Networks, we actually tried to intentionally excluded normalization part from them. However, we could not control the exploding gradients caused by the exclusion of normalization.
>
> - More data sets
> As described in Section 5, we did not experiment on ImageNet only because it requires much time to train each model although we need to train many models. We need to fine-tune the models with different regularization parameters, and we also need multiple training sessions of each model to obtain confidence interval. For example, the experiment results in Figure 3 and 4 include 750 training sessions in total. This is something we cannot afford with ImageNet data set, which requires several weeks of training for EACH session (unless you have GPU clusters). However, we instead performed more experiments on another data set. Specifically, we will add results of different VGG architectures on the SVHN data set, in order to see the difference in the tolerance level that is caused by a different number of hidden layers. We will add these results in the revised version.
>
> - Explanation of Figure 5
> Here is the detailed explanation of Figure 5. Through the grace period where the regularization parameter is zero, we expect the model to reach an "active learning" phase with an elevated gradient amount (e.g., green and blue lines in Figure 2b reach there in a couple of epochs). We hypothesize that once the model reaches there, it does not suffer from vanishing gradients any more even when strong regularization is enforced. We empirically show that the hypothesis is valid in Figure 5a, where the gradient amount does not decrease when the strong regularization is enforced (at epoch=5).
>
>  In Figure 5b, although the same strong regularization is enforced since epoch 5, the magnitude of weights in our model stops decreasing around epoch 20, while that in baseline keeps decreasing towards zero. This means that our model can cope with strong regularization, and it maintains its equilibrium between gradients from L and those from regularization. We will change the Figure 5b and its description to make it more clear.
>
> - p-value
> We did not compute p-values since we only ran three training sessions for each model. However, as suggested by the reviewer, we computed the p-value and found that most improvements are statistically significant (p < 0.05). We will include the exact p-values in the revised version.
>
> Thank you again for the comments, and we will make them clear in the next version.

---

> ### Author Response · Authors · 2018-01-03
> **The sudden failure in performance found by our analysis is novel.**
>
> We thank the reviewer for asking important questions.
>
> 1. Novelty
>
> We agree with the referee that the phenomenon that the performance first increases and then falls down as the regularization parameter increases is well known for constrained optimization problems. However, we observe a different phenomenon that the performance first increases and then "suddenly" fails at some point as the regularization parameter increases in deep networks. The sudden failure (as opposed to gradual falling) in performance found by our analysis is novel.
>
> Also, in order to claim that the sudden failure in performance also follows from vanishing gradients problem in deep networks, we would at least need to know that the gradients would SUDDENLY vanish as the regularization parameter increases. To the best of our knowledge, there is no such an analysis, be it theoretical or empirical. In contrast, we conduct our analysis in an opposite direction — first of all, we empirically demonstrated that the early-stage weights would diminish suddenly, when the regularization parameter is increased right above a certain threshold. This together with the aid of the derivation in Section 2.2 lead us to conclude that the gradients would suddenly vanish as the regularization parameter increases.
>
> In other words, knowing the empirically found diminishing weights, we have explained the sudden failure in performance by sudden vanishing gradients through the derivation in Section 2.2. Indeed, this intuition has also guided us to introduce the Delayed Strong Regularization in this paper. (Please see the difference between Delayed Strong Regularization and Reducing Regularization Parameter clarified below.)
>
>
>
> 1a. Delayed Strong Regularization vs. Reducing Regularization Parameter
>
> They are not equivalent. Reducing the regularization parameter means that we enforce weaker regularization in each training step. This is different from our approach (Delayed Strong Regularization) where we enforce the same strong regularization in each training step after five (\gamma) epochs. In fact, by skipping regularization for the first five epochs out of 300 epochs, the total reduced amount by regularization throughout training is decreased. However, the decreased amount is negligible. Indeed, our approach does not fail in learning with regularization parameter that is two orders of magnitude greater than the highest regularization parameter the baseline can adopt without a fail.
>
> In case the reviewer meant reducing the regularization strength by "gradually" reducing regularization strength throughout the training, we also performed a simple experiment with VGG-16 on CIFAR-100. We set the initial regularization parameter \lambda=2*10^-3 and 6*10^-5 for L2 and L1 regularization, respectively, which are just above the "tolerance level". Then, we continuously reduced \lambda to zero throughout the training session. The trained models didn't show any improvement over "random guess", which means that they were not able to learn.
>
>
>
> 1b. Why should we keep increasing the regularization constant beyond a limit?
>
> Often, deep neural networks need strong regularization especially when they are too complex while training data is small. Although data is key in deep learning, it is often very expensive to obtain, so it is difficult to secure enough data set for the networks in practice. When the model is overfitted, one possible solution is to keep increasing the regularization strength, and strong regularization may boost the accuracy of the networks.
>
> Although many models are still over-fitted, stronger regularization cannot be achieved due to the vanishing gradient problem in the deep networks, as described in the paper (especially Section 2.2). In the beginning of the training, where gradient is small for stochastic gradient descent method, we find that learning fails if strong regularization is enforced. However, we find that we can overcome this by waiting for the model to reach an "active learning" phase, where the gradients' magnitudes are significant, and then enforcing strong regularization. Delayed Strong Regularization enables us to obtain the superior performance that is otherwise hidden by learning failure in deep networks.
>
> Strong regularization provides not only a better accuracy for over-fitted models but also more model compression. We show that we can achieve 2 to 4 times more compression compared to the baseline. The model compression can be done by other approaches such as pruning and quantization, but compression by regularization is also effective especially for removing neurons in groups with group sparsity. Certainly, there are recent efforts on this direction (Wen et al., 2016; Scardapane et al., 2017; Yoon & Hwang, 2017). Although our approach is not applied to group sparsity regularization in this paper, our approach has no limit on it.

---

### Official Review · AnonReviewer1 · 2017-11-28
**An interesting diagnosis but a rudimentary cure**

**Rating:** 5
**Confidence:** 5

**Review:**

The work was prompted by  an interesting observation: a phase transition can be observed in deep learning with stochastic gradient descent and Tikhonov regularization. When the regularization parameter exceeds a (data-dependent) threshold, the parameters of the model are driven to zero, thereby preventing any learning. The authors then propose to moderate this problem by letting the regularization parameter to be zero for 5 to 10 epochs, and then applying the "strong" penalty parameter. In their experimental results, the phase transition is not observed anymore with their protocol. This leads to better performances, by using penalty parameters that would have prevent learning with the usual protocol.

The problem targeted is important, in the sense that it reveals that some of the difficulties related to non-convexity and the use of SGD that are often overlooked. The proposed protocol is reported to work well, but since it is really ad hoc, it fails to convince the reader that it provides the right solution to the problem. I would have found much more satisfactory to either address the initialization issue by a proper warm-start strategy, or to explore standard optimization tools such as constrained optimization (i.e. Ivanov regularization) , that could be for example implemented by stochastic projected gradient or barrier functions. I think that the problem would be better handled that way than with the proposed strategy, which seems to rely only on a rather limited amount of experiments, and which may prove to be inefficient when dealing with big databases.

To summarize, I believe that the paper addresses an important point, but that the tools advocated are really rudimentary compared with what has been already proposed elsewhere.

Details :
- there is a typo in the definition of the proximal operator in Eq. (9)
- there are many unsubstantiated speculations in the comments of the experimental section that do not add value to the paper
- the figure showing the evolution of the magnitude of parameters arrives too late and could be completed by the evolution of the data-fitting term of the training criterion

---

> ### Author Response · Authors · 2018-01-03
> **We tried other approaches, but ours achieved the best accuracy while it is the simplest.**
>
> We thank the reviewer for the insightful comments.
>
> We agree that our proposed approach is very simple. The reason we employed the simple method is that it is effective while it is simple to implement for readers. The only additional hyper-parameter, which is the number of initial epochs to skip regularization, is also not difficult to set. We think that the proposed method is very close to the traditional regularization method so that it inherits the traditional one's good performance for non-strong regularization while it also achieves strong regularization.
>
> We actually tried a couple more approaches other than the proposed one in our preliminary experiments. We found that the proposed one shows the best accuracy among the approaches we tried while it is the simplest. For example, we tried an approach that can be regarded as a warm-start strategy. It starts with the regularization parameter \lambda_t=0, and then it gradually increases \lambda_t to \lambda for \gamma epochs, where \gamma >= 0 and it is empirically set. We found that it can achieve strong regularization, but its best accuracy is slightly lower than that of our proposed approach. We think that this is because our model can explore the search space more freely without regularization while the warm-start model enforces some regularization during the warm-up stage.
>
> We also tried a method that is similar to Ivanov regularization. In this method, the regularization term is applied only when the L1 norm of the weights is greater than a certain threshold. To enforce strong regularization, we set the \lambda just above the tolerance level that is found by the baseline method. However, this method did not accomplish any learning. The reason is that, to reach the level of L1 norm that is low enough, the model needs to go through the strong regularization for the first few epochs, and the neurons already lose its learning ability during this period like the baseline method. If we set the lambda below the tolerance level, it cannot reach the desired L1 norm without strong regularization, and thus the performance is inferior to our proposed method.
>
> We did not extend these preliminary experiments to full experiments because the required number of training sessions is overwhelming, and the preliminary results were not promising. As mentioned in the answers to the other reviewers, the number of training sessions needed for the results in Figure 3 and 4 was 750, which takes quite much time. We will add this discussion to the paper in the new version to make it clear. We will also add more experiment results on another data set to convince readers of our proposed method's superiority. Specifically, we will add results of different VGG architectures on the SVHN data set, in order to see the difference in the tolerance level that is caused by a different number of hidden layers.
>
> We also thank the detailed comments at the end of the review. We agree with the reviewer, and we will revise the paper accordingly.

---

### Author Response · Authors · 2018-01-05
**A revised version of paper is uploaded.**

We made the following changes in the revised version:

- We did additional proofreading.

- We adjusted resolution of figures for better presentation.

- We added results of VGG variations for additional data set (SVHN) in Section 5 and Appendix A.

- We explained why we use simple method and what other methods we tried in Section 2.3 and Appendix B.

- We fixed a typo in proximal operator function in Section 2.3.

- We explained how our method is different from using slightly weaker regularization strength in Section 2.3.

- We added explanation of Figure 5.

- We added p-values for improvements.

- We removed unsubstantiated speculations in Section 3.

- We further explained why we did not use ImageNet data set in Section 5.

- We explained that our method could not be applied to ResNet without normalization in Section 5.

---

### Decision · Program_Chairs · 2018-01-29
**ICLR 2018 Conference Acceptance Decision**

**Decision:**

Reject

**Comment:**

The submission is motivated by an empirical observation of a phase transition when a sufficiently high L1 or L2 penalty on the weights is applied.  The proposed solution is to optimize for several epochs without the penalty followed by introduction of the penalty.  Although empirical results seem to moderately support this approach, there does not seem to be sufficient theoretical justification, and comparisons are missing.  Furthermore, the author response to reviewer concerns contain unclear statements e.g.
"The reason is that, to reach the level of L1 norm that is low enough, the model needs to go through the strong regularization for the first few epochs, and the neurons already lose its learning ability during this period like the baseline method."
It is not at all clear what "neurons already lose its learning ability" is supposed to mean.